# Mobile Phone in the Lives of Young People of Rural Mountainous Areas of Gilgit-Baltistan, Pakistan: Challenges and Opportunities

**Sabit Rahim** [1,*] **, Gul Sahar** [1] **, Gul Jabeen** [1] **, Akber Aman Shah** [2] **, Musrat Jahan** [1] **and Tehmina Bibi** [3]

1   Department of Computer Sciences, Karakoram International University, Gilgit-Baltistan 15100, Pakistan;
    gulsahar@kiu.edu.pk (G.S.); gul.jabeen@kiu.edu.pk (G.J.); jhn_msrt@yahoo.com (M.J.)
2   Department of Management and Business Administration, Rawalpindi Women University,
    Rawalpindi 46200, Pakistan; akberaman@gmail.com
3   Institute of Geology, University of Azad Jammu and Kashmir, Muzaffarabad 13100, Pakistan;
    tehmina.bibi@ajku.edu.pk
*   Correspondence: sabit.rahim@kiu.edu.pk

**Abstract:** This research aims to investigate the impact of mobile phones in the lives of youths of mountainous rural areas of Gilgit-Baltistan (GB). A total of 272 (133 male and 139 female) respondents of ages between 16 and 25 years participated in this study. To analyze the demographic data such as age, gender, district, the descriptive statistics (mean, SD and percentage) and inferential statistics such as independent sample $t$-test were used. The regression analysis was used to analyze the relationship between independent and dependent variables such as mobile phone features (M = 3.66, SD = 1.15); a mobile phone as a tool for socio-economic impact (M = 3.80, SD = 1.20); as a fashion symbol (M = 1.29, SD = 0.11) and a tool for safety (M = 3.91, SD = 1.06). The findings show that 97% (M = 1.026 SD = 0.159) of youths from GB own a mobile phone (47% male and 48% female). The findings also verify that a mobile phone is beneficial to its users in the fields of economic, education, safety, and security. However, using a mobile phone as status symbol could have a negative impact on the lives of youths. This study recommends that the government should develop effective and efficient policy for mobile phone usage and users should also be aware of the blessings and risks associated with using a mobile phone in their lives.

**Keywords:** mobile phone; safety and security; economic development; technology; mountainous-rural areas

## 1. Introduction

In this global village, mobile phone technologies have brought about a revolution in the lives of people, especially the young people belonging to rural mountainous areas of developing countries such as Pakistan. Mobile phones provide many key features, such as communication in real-time, access to information on an anytime anywhere basis, and portability [1,2]. These technologies have a profound impact on the way users play a vital role in society [3,4]. Young people in Gilgit-Baltistan (GB) use mobile phones as powerful technological devices in their lives for a variety of purposes. However, mobile phones have both positive and negative impacts in the lives of young people of GB. They have dramatically changed the approaches for communication, and youths have been found to interact in a different manner as compared to earlier ways of interacting [5]. Thus, mobile phone technology as a powerful gadget that has greatly impacted the lifestyle of youths who use this technology for multiple purposes, such as for learning, social networking, and entertainment, because it has become

the manifestation of the digital age [6,7]. Young people also use mobile phones to access and share relevant information, using text messaging services and communication portals [8,9].

Using a mobile phone rather than a fixed telephone line is a way to keep in touch with family, friends, colleagues and business associates [10]. It also provides tremendous business facilities such as the online ordering of products and transactions [11,12]. Mobile phones have become a social phenomenon all over the world especially in the least developed countries such as in Africa, Pakistan, India, Bangladesh and Iran [13,14]. A few years ago, mobile phones were only limited to a few cities. Due to their rapid penetration, mobile phone coverage has also covered most rural areas, such as GB, which has filled the gap between developed and underdeveloped areas and provides the benefits of the technology.

Almost 30% of the world's population uses a mobile phone, which has become a very common technological device [15]. Before the egress of mobile phone technology in Gilgit-Baltistan, the landline was the only source for communication [16]. Due to the hard access policy of landlines, people did not show their interest. In 2007, mobile phones were introduced in GB by the Special Communication Organization (SCO), and people, especially young users, took more interest due to their features, such as portability, privacy, low cost and being more accessible for rural people, especially for the youth.

The mobile phone has brought many changes and impacts in the lives of people of rural mountainous areas, especially in the youth, such as socio-economic impacts, a sense of security, educational, and psychological impacts. This study focuses on the access and use of mobile phones by the youth of the rural mountainous area in their daily lives, such as mobile phone features, mobile phones as a tool for socio-economic impact, as a fashion symbol and a tool for safety. The growth of mobile phones in GB among the youth is very fast and there is dearth of research studies regarding the impact of mobile phones in youths and teenagers and society at large. This study will fill the gap of access and usage of mobile phone among the young people of GB. The paper has been organized as follows. The literature review is in Section 2, Section 3 presents the objectives of the study, Section 4 contains the methodology, the findings are included in Section 5, the discussion is included in Section 6, the implications of study are in Section 7, and Section 8 includes the limitations and future direction of research, while Section 9 includes the conclusion.

## 2. Literature Review

### 2.1. Access and Use of Mobile Phone

Mobile phones, including basic phones (e.g., a simple device that is used to send text messages and voice calls, etc.), smart phones (e.g., a device that provides more sophisticated applications of technology in addition to a basic phone), and other types of phones have been used extensively all over the world by youths aged 16–25, and their exploitation increases day by day [9]. Initially, the mobile phone was used only for communication purposes [3]. The research findings in American college students revealed that among youths, social interaction with friends and family, safety, cost effectiveness, information access and privacy are the main reasons to access and use a mobile phone [4].

Young people use mobile phones for various purposes [17], for example entertainment, social networking, security, accessing online learning materials, web browsing, a status symbol [18–20] and economic activities [21]. The basic purpose of the mobile phone was to connect users with each other over thousands of miles [22]. The advancement of mobile phone technology, such as portability, enabled users to connect on an anytime anywhere basis [15,23,24]. Besides its basic purpose of communication, research shows that [25] the mobile phone has altered the prime concept of communication, such as enhancing facilities for health, education, disaster reduction, ensuring security for females, social interaction, and facilitates in travelling.

The previous studies revealed that most young people use mobile phones among their peers to show their popularity, send text messages, keep their peers' and family's contact numbers, and use for communication [26–28]. Research showed that a large number of young people are dependent

on mobile phone to keep contact with friends and family members via social network sites such as Twitter, Facebook, LinkedIn, and also consulting for lectures and using the Internet for learning purposes [3,29,30]. However, the unlimited access to mobile phones has many negative implications for students [31], such as the ringing of mobile phones during class, which affects the whole learning environment, while there are more chances to use mobile phones for cheating purposes during examinations [32].

### 2.2. Mobile Phone Usage Difference of Gender and Age Wise

Mobile phones have empowered both males and females equally in all aspects of life, such as use in business, security and learning purposes, which eliminates gaps between male and female income variations [33,34]. Thus, the mobile phone is a technological device that has equally influenced the lives of both the male and female segments of society in almost all districts of GB. Therefore, the access and use of mobiles in all aspect of life are a basic necessity for everyone, and people use them in effective and efficient way.

Mobile phones provide a variety of services, such as social networks [21], sharing of information with colleagues, safety and security for females, political activities and teaching and learning [13] among youths, especially girls at the university level. However, innovative things have a tendency to create anxieties among their users (i.e., a challenge that old people face, especially females; they feel uncomfortable about using mobile phones). However, there is a rapid growth in the adoption of mobile phones among both males and females and among different age groups; this has also brought about challenges for elder users [35]. The authors in [4] studied the behaviors of mobile phone users such as safety and cost consciousness, dependent users, sophisticated users and practical users.

The young people of GB use mobile phones in positive ways such as listening to audio and video lectures, sharing knowledge via social media or SMS, and accessing educational applications (like mobile dictionary) [36–39]. Students use mobile phone devices to upload and post their academic data to course websites and each student get access to their course [40,41]. At the same time, most of the mobile phone users use it as a status symbol [42].

There is positive attitude of mobile phone users, such as females who stated that they feel safe while travelling because they can easily inform their parent about their movement and contact them in an emergency [34]. The researchers divided mobile phone users into groups such as safety and sophisticated users. Whereby safety users, such as young females, reported that while travelling they feel safe and secure having mobile phone, the sophisticated users use mobile phones for social activities such as using social media, sending text messages, and contacting friends and family members [43].

### 2.3. The Economic Impact of Mobile Phone on Youth

Young people especially, and all users, use mobile phones for not only communication purposes but also for promoting business, entertainment, teaching and learning and for safety and security purposes [2]. In a study held in rural Bangladesh, regarding the use of mobile phones, 74.2% respondents said that mobile phones had decreased the cost of communication, while 85% believed that overall communication cost has been cut down and 79.2% said that it also reduced the travel cost [44,45]. In [46], the authors revealed that Australian students in their study had an average bill of AUD140, which is considered very high for low income groups. Youths often use mobile phone while driving, which is a considerable safety risk [47] and has negative economic implications for their families.

The mobile phone is one of the technologies which releases users from the restriction of space and time, but at the same time, it has deteriorated the present limitation of social norms [48]. Previous research showed that mobile phones not only have an impact on economic factors but also have a negative impact on young people's norms and values in the society. The researchers in [49] showed that students use mobile phones for social networking and entertainment, which has economic implications. Overall, 92% said that they use mobile phones for sending text messages, while 86% said that they use them to check emails. Most young people use mobile phones' latest features for getting information

and sharing knowledge [50,51], which shows that they are an essential component for college and university students [52].

Most young people in rural areas consider mobile phones as a status symbol as part of how they dress and align them with their lifestyle, status position and sense of fashion [53]. Most of the youth considers style as an important component while purchasing a mobile phone and they do not even consider their buying power.

Despite many benefits, there is a cost problem regarding mobile phone purchase and daily expenses due to the excessive use of mobile phones among young people, which increases the economic burden on users [47].

## 3. Objectives of the Study

The objectives of this study are as follows:

- The access to mobile phones by youth across the eight districts of Gilgit-Baltistan Province [3,40,41].
- The differences in the usage patterns of mobile phones between males and females across eight districts of Gilgit-Baltistan [34,53].
- The economic impact of mobile phone us by the youth of GB province, compared by gender [46,47]

## 4. Methodology

### 4.1. Theoretical Framework

A theoretical framework was developed for the current research based on the literature review which is a guideline and a conceptual map for this research (Figure 1). Four major categories were identified, such as mobile phone (MP) features, the use of mobile phones for safety, use as a fashion symbol and mobile phones' impact on economic factors [3,4,13,18–21].

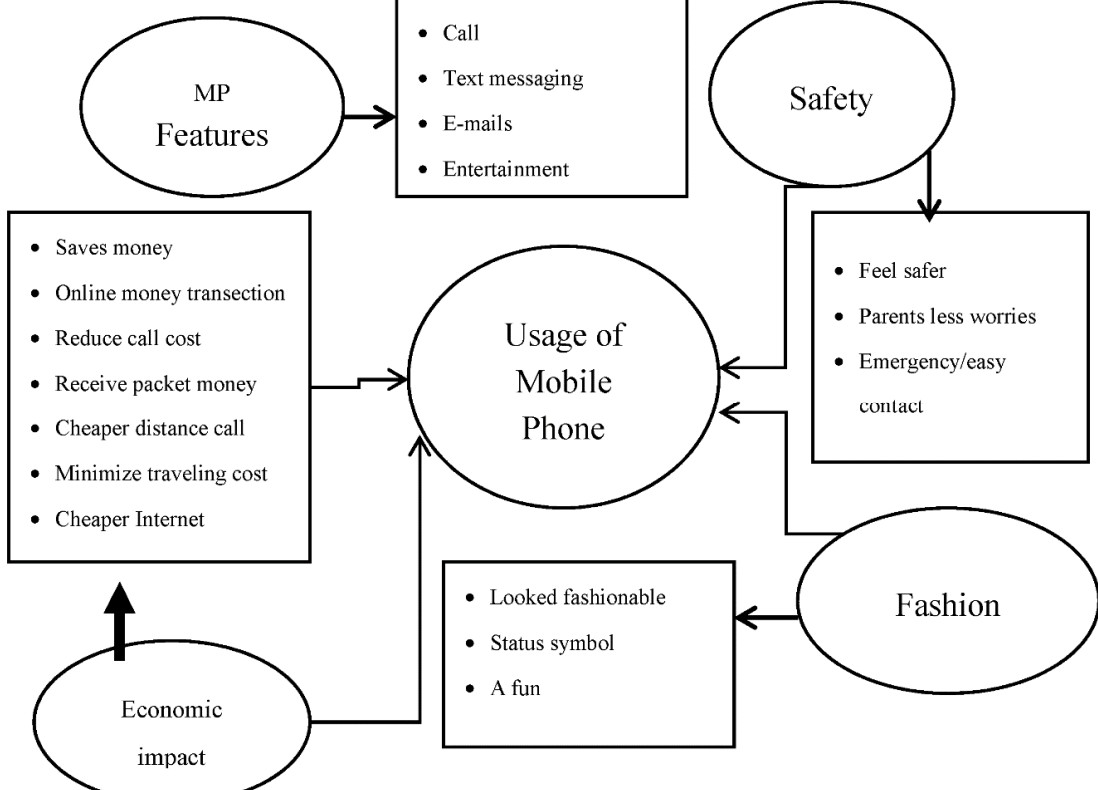

**Figure 1.** Theoretical framework.

*4.2. Sample and Procedure*

This study was carried out in Gilgit-Baltistan (GB), Pakistan; 272 young people aged 16–26 participated in this survey. The participants were selected with the procedure of random sampling from eight districts of GB, from which 139 female and 133 male participants showed their interest; only 8 participants did not respond the questionnaires (Table 1). The 29-item questionnaires consisted of two major sections. Section A consisted of two sub-sections: the first sub-section contained demographic information such as participants' age, gender, and district of origin, and the second sub-section related to information about access to a mobile phone, types of mobile phone (simple or smart), and spending money on a mobile phone (balance card, easy load) per month in Pakistani rupees (assessed by 1 equal to Yes and 0 equal to No). Before the distribution of questionnaires among the participations, five students were asked to review the questionnaires for the better understanding of questionnaires by university students. A few questions were rephrased. Reliability tests were also performed, and the result of the reliability test shows a Cronbach's alpha value of 0.841, which is an acceptable level of internal consistency.

The questions on access, kinds (simple or smart phone) and the amount (in Pakistani rupees) spent monthly on mobile phones was also asked from respondents (Table 2).

Section B: This section further divided into four parts, such as mobile phones as a fashion symbol, mobile phone features, mobile phones for safety purposes and the economic impact of mobile phones on the youth of GB, as shown in Table 3.

**Table 1.** Participants characteristics (respondents age, gender, and income).

| Gender | Age | Ghizer | | Gilgit | | Hunza | | Nagar | | Skardu | | Ghanche | | Astore | | Diamer | | No Response | |
|---|---|---|---|---|---|---|---|---|---|---|---|---|---|---|---|---|---|---|---|
| | | *N* | % | *N* | % | *N* | % | *N* | % | *N* | % | *N* | % | *N* | % | *N* | % | *N* | % |
| | 16–18 | 2 | 1 | 12 | 4 | 2 | 1 | 2 | 1 | 0 | 0 | 3 | 1 | 2 | 1 | 0 | 0 | 0 | 0 |
| Male | 19–21 | 19 | 7 | 21 | 8 | 9 | 3 | 11 | 4 | 18 | 7 | 2 | 1 | 7 | 3 | 10 | 4 | 3 | 1 |
| | 22–25 | 2 | 1 | 2 | 1 | 2 | 1 | 0 | 0 | 1 | 0 | 1 | 0 | 0 | 0 | 2 | 1 | 0 | 0 |
| | 16–18 | 4 | 1 | 6 | 2 | 5 | 2 | 3 | 1 | 3 | 1 | 0 | 0 | 3 | 1 | 0 | 0 | 2 | 1 |
| Female | 19–21 | 22 | 8 | 18 | 7 | 29 | 11 | 8 | 3 | 17 | 6 | 4 | 1 | 8 | 3 | 1 | 0 | 1 | 0 |
| | 22–25 | 2 | 1 | 1 | 0 | 0 | 0 | 0 | 0 | 0 | 0 | 2 | 1 | 0 | 0 | 0 | 0 | 0 | 0 |
| | Income of Respondents | | | | | | | | | | | | | | | | | | |
| | Personal | 1 | 0 | 6 | 2 | 0 | 0 | 2 | 1 | 3 | 1 | 0 | 0 | 0 | 0 | 1 | 0 | 0 | 0 |
| Male/Female | Family | 49 | 18 | 54 | 20 | 47 | 17 | 22 | 8 | 35 | 13 | 12 | 4 | 20 | 7 | 12 | 4 | 6 | 2 |
| | No response | 1 | 0 | 0 | 0 | 0 | 0 | 0 | 0 | 1 | 0 | 0 | 0 | 0 | 0 | 0 | 0 | 0 | 0 |

**Table 2.** Mobile phone access, types and money spent on mobile phones monthly.

| Items | Variables | Ghizer | | Gilgit | | Hunza | | Nagar | | Skardu | | Ghanche | | Astore | | Diamer | | No Response | |
|---|---|---|---|---|---|---|---|---|---|---|---|---|---|---|---|---|---|---|---|
| | | *N* | % | *N* | % | *N* | % | *N* | % | *N* | % | *N* | % | *N* | % | *N* | % | *N* | % |
| | Access of Mobile Phone | | | | | | | | | | | | | | | | | | |
| Yes | | 49 | 18 | 58 | 21 | 45 | 17 | 24 | 9 | 38 | 14 | 12 | 4 | 20 | 7 | 13 | 5 | 6 | 2 |
| No | | 2 | 1 | 2 | 1 | 2 | 1 | 0 | 0 | 1 | 0 | 0 | 0 | 0 | 0 | 0 | 0 | 0 | 0 |
| | Which kind of mobile phone do you have? | | | | | | | | | | | | | | | | | | |
| Simple mobile phone | | 22 | 8 | 26 | 10 | 17 | 6 | 8 | 3 | 16 | 6 | 1 | 0 | 10 | 4 | 7 | 3 | 2 | 1 |
| Smart mobile phone | | 17 | 6 | 18 | 7 | 18 | 7 | 8 | 3 | 14 | 5 | 6 | 2 | 6 | 2 | 5 | 2 | 1 | 0 |
| Both simple and smart mobile phone | | 10 | 4 | 14 | 5 | 10 | 4 | 8 | 3 | 8 | 3 | 5 | 2 | 4 | 1 | 1 | 0 | 2 | 1 |
| No response | | 2 | 1 | 2 | 1 | 2 | 1 | 0 | 0 | 1 | 0 | 0 | 0 | 0 | 0 | 0 | 0 | 1 | 0 |
| | Spending rupees on mobile phone (balance card, easy load) per month | | | | | | | | | | | | | | | | | | |
| Less than 100 | | 0 | 0 | 1 | 0 | 0 | 0 | 0 | 0 | 3 | 1 | 1 | 0 | 2 | 1 | 0 | 0 | 0 | 0 |
| 100–200 | | 1 | 0 | 3 | 1 | 1 | 0 | 1 | 0 | 1 | 0 | 0 | 0 | 1 | 0 | 0 | 0 | 0 | 0 |
| 200–500 | | 11 | 4 | 14 | 5 | 19 | 7 | 4 | 1 | 8 | 3 | 2 | 1 | 4 | 1 | 0 | 0 | 0 | 0 |
| Above 500 | | 16 | 6 | 26 | 10 | 22 | 8 | 7 | 3 | 11 | 4 | 3 | 1 | 9 | 3 | 5 | 2 | 1 | 0 |
| No response | | 30 | 11 | 23 | 8 | 21 | 8 | 14 | 5 | 23 | 8 | 7 | 3 | 8 | 3 | 8 | 3 | 5 | 2 |

**Table 3.** Mobile phone characteristics.

| Mobile Phone Use for Fashion | Strongly Disagree | | Disagree | | Somehow | | Agree | | Strongly Agree | | No Response | |
|---|---|---|---|---|---|---|---|---|---|---|---|---|
| | N | % | N | % | N | % | N | % | N | % | N | % |
| I use mobile phone to be looked fashionable | 92 | 33.8 | 83 | 30.5 | 37 | 13.6 | 45 | 16.5 | 11 | 4.0 | 4 | 1.5 |
| Mobile phone is a status symbol | 53 | 19.5 | 48 | 17.6 | 46 | 16.9 | 105 | 38.6 | 17 | 6.3 | 3 | 1.1 |
| I use mobile phone because it is a fun | 65 | 23.9 | 79 | 29 | 52 | 19.1 | 58 | 21.3 | 14 | 5.1 | 4 | 1.5 |
| **Features of Mobile Phone** | | | | | | | | | | | | |
| I use mobile phone mostly for calling | 12 | 4.4 | 22 | 8.1 | 62 | 22.8 | 106 | 39 | 64 | 23.5 | 6 | 2.2 |
| I use it for text messaging | 2 | 0.7 | 19 | 7 | 32 | 11.8 | 126 | 46.3 | 87 | 32 | 6 | 2.2 |
| I use it for sending e-mails | 28 | 10.3 | 56 | 20.6 | 62 | 22.8 | 79 | 29 | 40 | 14.7 | 7 | 2.6 |
| I use it for entertainment | 26 | 9.6 | 33 | 12.1 | 59 | 21.7 | 85 | 31.3 | 62 | 22.8 | 7 | 2.6 |
| **Mobile Phone Safety purposes** | | | | | | | | | | | | |
| Having mobile phone, I feel safer | 13 | 4.8 | 40 | 14.7 | 59 | 21.7 | 96 | 35.3 | 58 | 21.3 | 6 | 2.2 |
| Having mobile phone Parents worries about me less | 11 | 4 | 21 | 7.7 | 37 | 13.6 | 130 | 47.8 | 67 | 24.6 | 6 | 2.2 |
| I use mobile phone to contact in an emergency | 7 | 2.6 | 8 | 2.9 | 23 | 8.5 | 109 | 40.1 | 119 | 43.8 | 6 | 2.2 |
| **Economic Impact of Mobile Phone** | | | | | | | | | | | | |
| It saves money which I spend on physical dictionary and wristwatch | 18 | 6.6 | 30 | 11 | 48 | 17.6 | 85 | 31.3 | 82 | 30.1 | 9 | 3.3 |
| I send and receive money online using mobile phone (easy paisa etc.) | 30 | 11 | 32 | 11.8 | 44 | 16.2 | 106 | 39 | 51 | 18.8 | 9 | 3.3 |
| Mobile phone has reduced the cost of communication | 7 | 2.6 | 16 | 5.9 | 48 | 17.6 | 106 | 39 | 88 | 32.4 | 7 | 2.6 |
| I receive my monthly pocket money using mobile phone | 35 | 12.9 | 59 | 21.7 | 45 | 16.5 | 77 | 28.3 | 47 | 17.3 | 9 | 3.3 |
| Mobile phone is Cheaper alternative for distance calling | 11 | 4 | 17 | 6.3 | 43 | 15.8 | 115 | 42.3 | 78 | 28.7 | 8 | 2.9 |
| Mobile phone has Minimized the travelling cost | 16 | 5.9 | 22 | 8.1 | 29 | 10.7 | 96 | 35.3 | 102 | 37.5 | 7 | 2.6 |
| Mobile Internet is less expensive than landline internet | 16 | 5.9 | 9 | 3.3 | 34 | 12.5 | 96 | 35.3 | 108 | 39.7 | 9 | 3.3 |

## 5. Findings

For data analysis, Statistical Package of the Social Sciences (SPSS 21.0) was used. The descriptive statistics such as mean, standard deviation, percentage, and inferential statistics such as independent *t*-test were applied for Sections A and B. The regression analysis was used to analyze the relationship between independent and dependent variables such as mobile phone use for fashion purposes, features of mobile phones, and their use for safety purposes.

### 5.1. Demographic

Two hundred and eighty participants were selected based on a random selection procedure; 272 responded from eight districts of GB, and only 8 respondents did not participate. The average age of participants was 21 years, ranging from 16 to 25. The participants' age groups are shown in Figure 2.

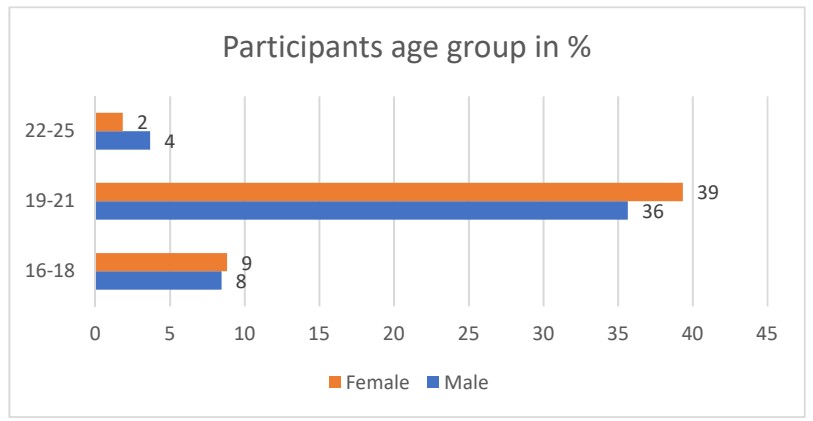

**Figure 2.** Sex-wise participants' age groups.

For the question asked regarding personal or family income, 5% participants responded that they have their own income sources (online earning), 94% responded that they have family income and 1% did not respond to the question (Table 1). The participants from the eight districts are shown in Figure 3.

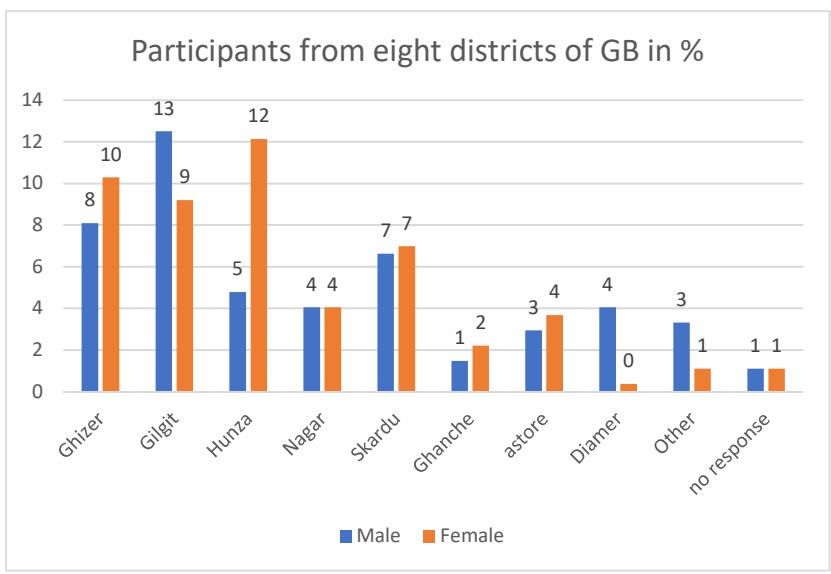

**Figure 3.** District-wise participants in percentage.

The participants were also asked about their access to a mobile phone, and 95% (47% male and 48% female) of the youths said that they own a mobile phone. The participants' responses regarding the kinds of mobile phone and how much rupees they spend on their mobile phone on a monthly basis are shown in Table 2.

*5.2. Usage of Mobile Phone*

Mobile Phone Use for Fashion

Figure 4 and Table 3 show the responses to mobile phones as a fashion symbol among the young people of GB. The *p*-value in Table 4 shows that there is no significant difference between male and female use of mobile phones for fashion purposes.

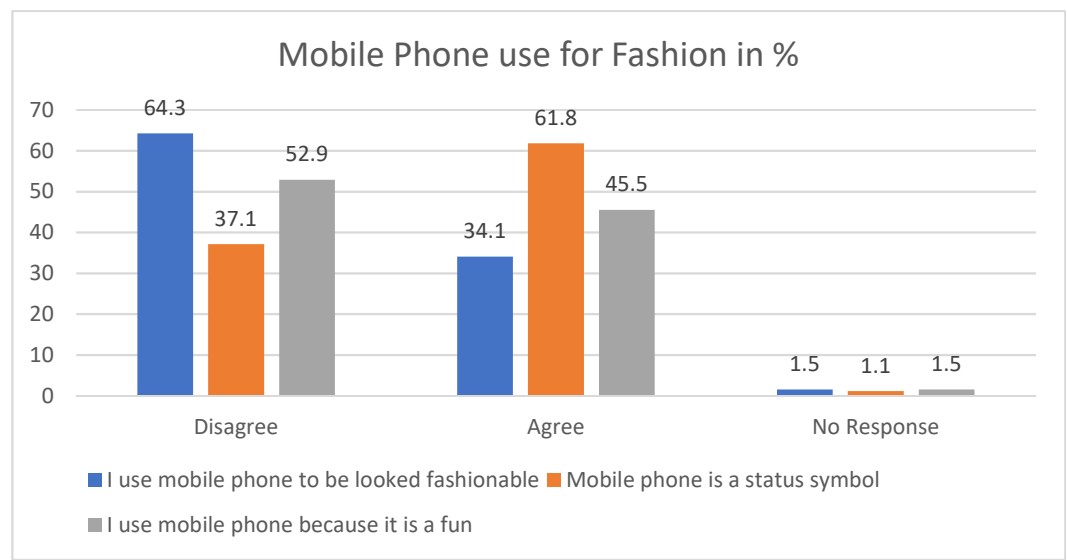

**Figure 4.** Mobile phone as a fashion symbol among young people.

**Table 4.** *T*-test for mobile phone use for fashion gender-wise.

| Items | Gender | *t*-Value | *p*-Value | Mean | Std. Deviation |
|---|---|---|---|---|---|
| Use of mobile phone as a status symbol | Male | 1.21 | 0.23 | 3.08 | 1.26 |
| | Female | | | 2.88 | 1.34 |
| I use a mobile phone to look fashionable | Male | 1.22 | 0.22 | 2.41 | 1.25 |
| | Female | | | 2.22 | 1.31 |
| I use a mobile phone because it is fun | Male | 1.07 | 0.29 | 2.68 | 1.25 |
| | Female | | | 2.51 | 1.30 |

*5.3. Use of Mobile Phone Features*

Figure 5 and Table 3 show the results regarding the features of mobile phone use by the young people of GB in their daily life activities.

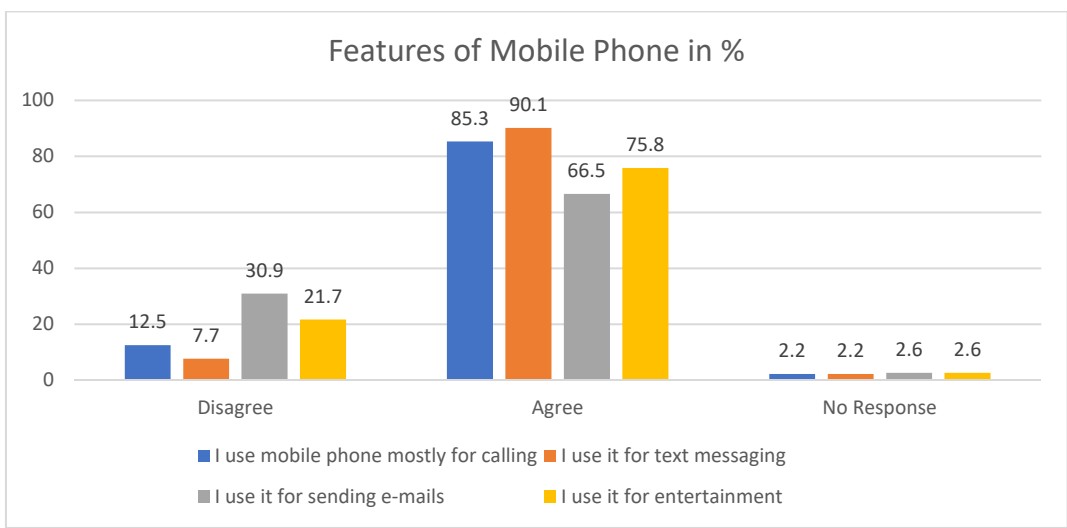

**Figure 5.** Mobile phone features used by young people in daily life activities.

The mean and standard deviation are shown in Table 5. The *p*-values are not significant except for the last item (I use it for entertainment) *p* = 0.04, which is less than 0.05. The values of other items are greater than *p* = 0.05; this shows that there is no significant difference in mobile phone usage between males and females, and both use mobile phone features in an equal ratio.

**Table 5.** *T*-test for use of mobile phone feature.

| Items | Gender | *t*-Value | *p*-Value | Mean | Std. Deviation |
|---|---|---|---|---|---|
| I use my mobile phone mostly for calling | Male | 0.91 | 0.36 | 3.82 | 1.06 |
| | Female | | | 3.70 | 1.15 |
| I use it for text messaging | Male | −0.42 | 0.67 | 4.06 | 0.89 |
| | Female | | | 4.11 | 0.97 |
| I use it for sending e-mails | Male | −0.87 | 0.39 | 3.18 | 1.25 |
| | Female | | | 3.32 | 1.33 |
| I use it for entertainment | Male | 2.08 | 0.04 | 3.70 | 1.27 |
| | Female | | | 3.37 | 1.31 |

### 5.4. Mobile Phone Use for Safety Purpose

In this part, Figure 6 and Table 3 show the results regarding the use of mobile phones for safety purposes by the young people of GB. The *t*-test result shows in Table 6 that there is no significant difference in the use of mobile phones for safety purposes between males and females. The mean values and standard deviation of all features of mobile phone use for safety are also shown in Table 6.

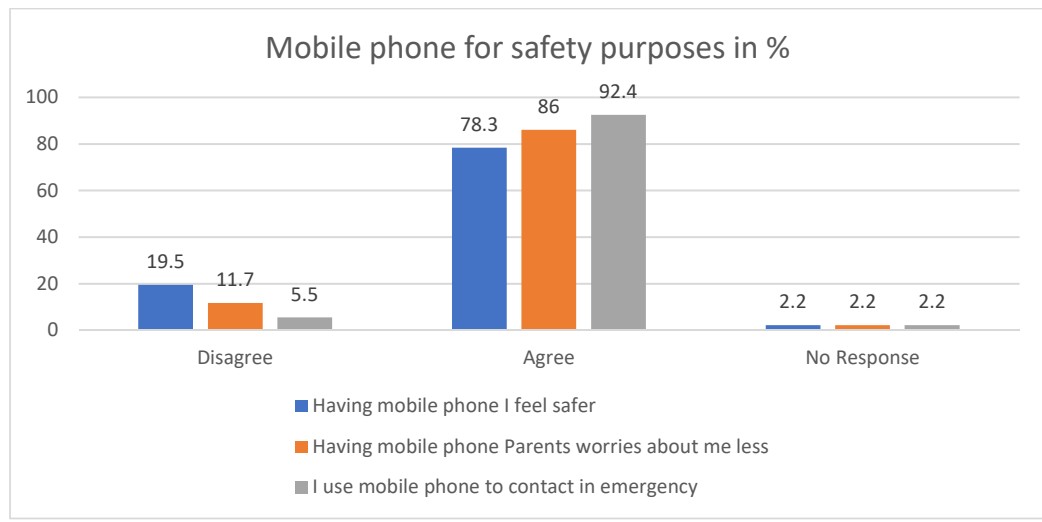

**Figure 6.** Use of mobile for safety purpose by young People.

**Table 6.** *T*-test of using mobile phone safety purposes.

| Items | Gender | *t*-Value | *p*-Value | Mean | Std. Deviation |
|---|---|---|---|---|---|
| Having cell phone, I feel safer | Male | −1.47 | 0.14 | 3.50 | 1.13 |
|  | Female | −1.47 | 0.14 | 3.71 | 1.21 |
| Having cell phone means that my parents worry about me less | Male | −1.58 | 0.11 | 3.77 | 0.97 |
|  | Female | −1.59 | 0.11 | 3.98 | 1.15 |
| I use my cell phone to contact in an emergency | Male | −1.25 | 0.21 | 4.19 | 0.90 |
|  | Female | −1.25 | 0.21 | 4.33 | 0.99 |

*5.5. Economic Impact of Mobile Phone*

Figure 7 and Table 3 show the results regarding the economic impact of mobile phones in the young people of GB. There is no significant difference in the use of mobiles for economic purpose by both males and females because the *p*-value is greater than 0.05, which is shown in Table 7.

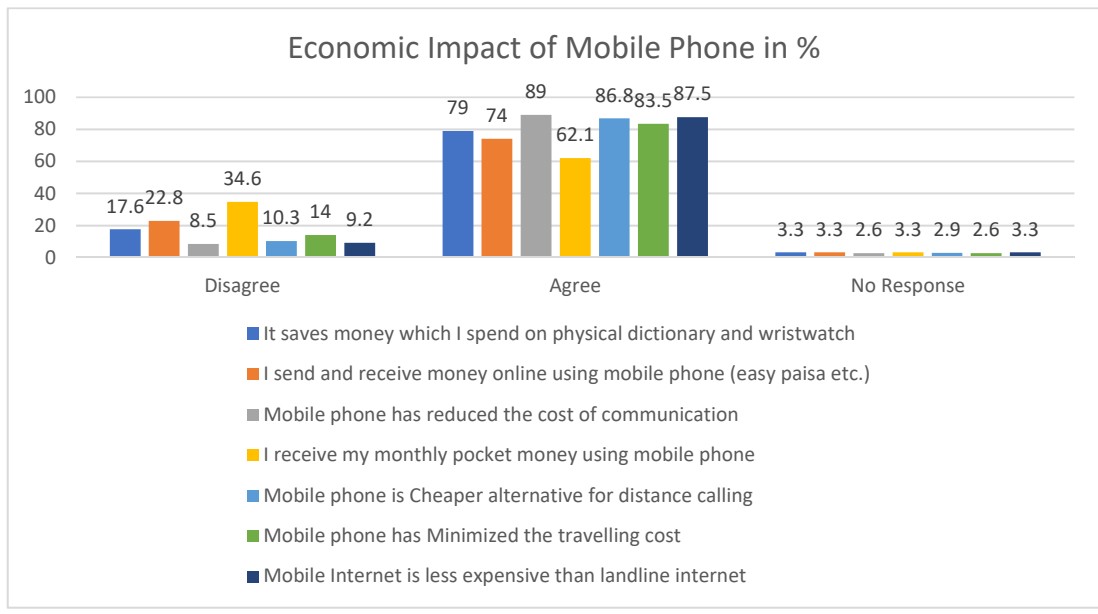

**Figure 7.** Economic impact of mobile phone use in the lives of young people of GB.

**Table 7.** *T*-test mobile phones' impact on the economic status of youths.

|  |  | *t*-Value | *p*-Value | Mean | Std. Deviation |
|---|---|---|---|---|---|
| It saves money which I spend on a physical dictionary and wristwatch | Male | −0.26 | 0.80 | 3.75 | 1.17 |
|  | Female |  |  | 3.79 | 1.35 |
| I send and receive money online using mobile (easy paisa etc.) | Male | 0.65 | 0.51 | 3.58 | 1.16 |
|  | Female |  |  | 3.47 | 1.45 |
| Mobile phone has reduced the cost of communication | Male | 0.18 | 0.86 | 4.02 | 0.86 |
|  | Female |  |  | 3.99 | 1.18 |
| I receive my monthly pocket money using mobile phone | Male | 1.77 | 0.08 | 3.41 | 1.34 |
|  | Female |  |  | 3.11 | 1.44 |
| Mobile phone is cheaper alternative for distance calling | Male | 0.87 | 0.38 | 4.00 | 0.93 |
|  | Female |  |  | 3.88 | 1.22 |
| Mobile phone has minimized the travelling cost | Male | 0.75 | 0.45 | 4.04 | 1.08 |
|  | Female |  |  | 3.93 | 1.31 |
| Mobile Internet is less expensive than landline internet | Male | 0.56 | 0.58 | 4.14 | 1.07 |
|  | Female |  |  | 4.06 | 1.21 |

The coefficient of mobile phone usage as a fashion symbol is 0.223, that of mobile phone features usage is 0.286 and that of mobile phone use for safety purpose is 0.514. However, the *t*-value as well as *p*-value clearly indicate that these variables are statistically significant because their *p*-values are less than 0.05. This means that a 1 percent change in the fashion, features, and safety leads to increase in the economic impact by 0.223, 0.268 and 0.514 percent, respectively. This result shows that the use of the mobile phone as a fashion symbol, its features, and its use for safety purposes have a great impact on the economic status of the youth of GB, increasing the expenses of youths who use a mobile phone (Table 8).

**Table 8.** Dependent variable economic impact.

|  | Coefficient | Std. Error | *t*-Statistics | *p*-Value |
|---|---|---|---|---|
| Constant | 9.315 | 1.545 | 6.030 | 0.000 |
| Fashion | 0.223 | 0.091 | 2.455 | 0.015 |
| Features | 0.286 | 0.089 | 3.215 | 0.001 |
| Safety | 0.514 | 0.113 | 4.537 | 0.000 |

As fashion, features, and safety increase, this directly influences the economy of the people, such as an increase in expenses of the youth. In the other hand, the mobile phone business will also increase at the same ratio as the increase in the expenses of youth.

## 6. Discussion

This study focused on the access of mobile phones in rural mountainous areas of GB, Pakistan. The sample comprised 272 young people of eight districts of GB, and all of them completed 29-item questionnaire. Four dimensions of the mobile phone usage pattern were focused on: the mobile phone as a fashion symbol, the use of a mobile phone's available features in a proper way, its use for safety purposes and mobile phone usage's impact on the economy of youths and parents. This study suggested a number of important conclusions.

The first conclusion is that mobile phones are used as a fashion symbol in rural mountainous areas' youth (as a status symbol, looking fashionable and as a fun gadget) [42]. The result reveals that both males and females have the perception that others use mobile phones as a status symbol, while when asking them about mobile phones as a fashion symbol, the answer was quite different, as 61% of the youths disagreed with the statement that they use it as a fashion symbol [53]. The statement shows that the rural youth use mobile phones in a positive way, but there is still a need for policy from government to provide effective utilization of mobile phones for teaching, learning and earning [21].

There is a fear that mobile apps and online games will grip young people very rapidly, which will ruin the lives of young people, which needs a strong and efficient policy.

The second conclusion is that mobile phones have made communication easy for rural mountainous areas, especially for youths worldwide. The majority of respondents use mobile phone for communication with parents, peers and friends (call, SMS, email and entertainment) [18–20]. There is no significant difference in the use of mobile phones for communication between males and females in the rural mountainous areas.

The third conclusion is that both males and females utilize mobile phone for safety purposes (feel safer, parents worry less about them and contact in an emergency) [47]. Due to current security threats of Pakistan, most of the parents worry about their children, both males and females; that is why 97% of young people own a mobile phone and positively use it for safety. They regularly inform their parents about their whereabouts.

The fourth conclusion is economic impact in the youth; the mobile phone is affordable in youths' daily lives as it is an important piece of technology in the current situation [21]. The effective and efficient utilization of the mobile phone saves money in many ways, such as providing access to software of learning materials (dictionary), the online transfer and receiving of money, and reducing long distance call cost, while mobile phone Internet has made it easy to access information on a anywhere and anytime basis. The utilization of mobile phones for safety, as a fashion symbol and its features increases the financial burden on the youth and their parents, but it also increases the economic benefits and increases the demand for the latest mobile phones [54].

## 7. Implication of Study

This study is one of the initiatives towards exploring access to and the use of mobile phones by the young people [55] of GB. It has many implications for future researchers as well as for readers (especially young boys and girls) in the context of GB. The area is strategically very important because of the China–Pakistan economic corridor which provides opportunities and challenges for the youth, and the country needs to plan and train young people accordingly. This study will also contribute in reducing one of the risk factors of young people using mobile phones to show their social status, leading them to suffer not only financially but also psychologically [56,57]. The study revealed that young people use mobile phones for learning, entertainment, safety, and security (females especially), communication, and economic benefits, which is a positive sign for rural mountainous people. Moreover, the use of mobile phones as fashion symbols, the use of emerging features of mobile phones (the Internet, apps, and social media) [21], and safety purposes directly influences the economy of the people, such as increasing the expenses of the youth. In the other hand, the mobile phone also provides emerging facilities such as online teaching and learning [13], sharing information, communication with friends and family, and allowing contact with relevant authorities in an emergency [4].

## 8. Limitation and Future Direction of Research

This research provides some important empirical evidence which proves that young people use mobile phones for different purposes such as a fashion symbol, for safety, and for economic purposes [58]. The study has a few limitations, such as being focused on mobile phone usage among young people only [59]; other users such as professionals and businesspersons who use similar devices extensively are not included in this study [60]. Other users such as professionals and businessmen and other devices such as iPads, tablet PCs, palm devices and laptops will be examined in future studies. The device's vital role in education, business and social relations will also be included in studies in future. The context of the mobile phone in business, and professional development, psychological issues and its negative impact on young people will also be included in future studies [61].

## 9. Conclusions

This study aimed to explore access to and use of mobile phones by the youth of mountainous rural areas of GB and their impact on their lives. To this end, the study focused on the level of mobile phone access, its use for fashion, the features of mobile phone use in daily life activities, mobile phone usage for safety purposes, and mobile phone usage's impact on the economic status of the young people of GB. The study concludes that there is no significant difference in the access to and use of mobile phones between males and females of rural mountainous areas of GB. One of the main findings of the study shows that the young people of GB tend to use mobile phones in a very positive manner for a variety of purposes such as access to knowledge, communicating with family and friends, use for safety and security purposes while travelling, and use for economic purposes. The findings show that 97% of young people in GB have access to a mobile phone and use them for a variety of purposes, such as entertainment, social networking, contacting peers and family, reducing economic burden and sharing information with others. Similarly, 78.3% of youths, both males and females, reported that their mobile phone has increased the feeling of safety in their families, which is a positive use of phones. The findings reveal that access to mobile phones created many opportunities and as well as many challenges. It is believed that positive use of technology (i.e., conscious use of technology with great awareness) can be a blessing, whereas its negative use can add risks in the lives of young people.

**Author Contributions:** Conceptualization, S.R., G.J., A.A.S., M.J. and G.S.; methodology, G.J. and M.J.; formal analysis, A.A.S. and M.J.; data collection, S.R., G.J. and M.J.; writing—original draft preparation, A.A.S. and G.S.; writing—review and editing, S.R. and T.B.; supervision and project administration, S.R. All authors have read and agreed to the published version of the manuscript.

**Funding:** This research received no external funding.

**Acknowledgments:** We would like to thank all participants who participated voluntarily in this survey.

**Conflicts of Interest:** The authors declare no conflict of interest.

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
