# Peer review of "Mobile Phone in the Lives of Young People of Rural Mountainous Areas of Gilgit-Baltistan, Pakistan: Challenges and Opportunities"

_information, doi:10.3390/info11090441_

Round 1
Reviewer 1 Report
This article aims to investigate the impact of mobile phones in the lives of youth from mountainous rural areas of Gilgit-Baltistan (G-B) in Pakistan. This kind of research is important to understand the impact of technology on the population, especially in young people living in rural areas.
The article is very interesting but it does not provide background information to understand when it was the mobile phone introduced in the rural area of G-B or what percentage of the young population has or does not have a mobile phone. As far as I understand many of the conclusions found by this research are also concluded by other similar investigations carried out in other countries or populations. Although the authors mentioned that this kind of research has not been performed in G-B, they should offer more background information about why they think the conclusions obtained from other populations may not be right for G-B or rural areas of Pakistan.
Author Response
I would like to thank to reviewer and editor for their time to review manuscript. Please find the reviewer comments and revised version
Similarity Index
Point 1: Similarity index of 1,2,4,8
Response 1: Similarity index of abstract, 1,2,4,5 and 8 has been revised
Response to Reviewer 1 Comments
Point 1: Under the English language and style. The English language and style are fine/minor spell check required
Response 1: The spellings have been checked and corrected in the following sections: abstract, introduction, literature review, discussion, limitation, and conclusion,
Point 2: Are the results clearly presented? Can be improved
Response 2: the results has been improved and figures have been added in the finding section.
Point 3: Are the conclusions supported by the results?
Response 3: The conclusion section has been aligned with results.
Comments and Suggestions portion
Point 1: background information to understand when it was the mobile phone introduced in the rural area of G-B
Response 1: The background information of mobile phone introduced in GB has been added in introduction section

Reviewer 2 Report
The paper presents the study on the use of mobile phones by young people in Rural Mountainous Areas of Gilgit-Baltistan, Pakistan. It shows the results of this study and the analysis.
The paper is well organized. The topic is interesting and the study was presented in interesting way. The study from the quite small area of one country in the world could be seen as its drawback but also it could be its advantage because it presents the topic from on special point of view.
The weak points of the paper are as follows:
- the description of paper structure has to be given at the end of the introduction
- the contribution of the paper has to be clearly stated
- the charts presenting different points of view of the given study will be very useful
- conclusion could be longer and detail
- discussion should presents in detail different points of view of the presented research
Author Response
I would like to thank reviewer for time to reviewe manuscript.
Response to Reviewer 2 Comments
Point 1: Does the introduction provide sufficient background and include all relevant references?
Response 1: The background information relevant to research has been included in the introduction section
Point 2: Is the research design appropriate?
Response 2: Little addition has been done in research design which was missed during writing the manuscript.
Point 3: Are the methods adequately described?
Response 3: The method section has also been improved.
Point 4: Are the results clearly presented? Can be improved
Response 4: The results has been improved and figures have been added in the finding section.
Point 5: Are the conclusions supported by the results?
Response 5: The conclusion section has been aligned with results.
Improvement on the weak points of the paper
The study has been taken in small area which is strategically very important because of China Pakistan economic corridor which may provide opportunities and challenges as well. In implication the sentences has been added.
Point 1: the description of paper structure has to be given at the end of the introduction
Response 1: The background information of mobile phone introduced in GB has been added in introduction section
Point 2: the contribution of the paper has to be clearly stated
Response 2: The contribution of paper has been clearly mentioned in the implication of study section.
Point 3: the charts presenting different points of view of the given study will be very useful
Response 3: in the finding section charts haven been added to present clear points of view about study
Point 4: conclusion could be longer and detail
Response 4: More detail has been added in conclusion section to align with the study
Point 5: discussion should present in detail different points of view of the presented research
Response 5: four main findings has been discussed in the discussion section and revised it and presented different point of view on present research.
